# STRILL: Phase I Trial Evaluating Stereotactic Body Radiotherapy (SBRT) Dose Escalation for Re-Irradiation of Inoperable Peripheral Lung Lesions

**DOI:** 10.3390/diseases12070153

**Published:** 2024-07-12

**Authors:** Davide Franceschini, Mauro Loi, Antonio Marco Marzo, Luca Dominici, Ruggero Spoto, Anna Bertolini, Lorenzo Lo Faro, Francesco La Fauci, Beatrice Marini, Luciana Di Cristina, Marta Scorsetti

**Affiliations:** 1Department of Radiotherapy and Radiosurgery, IRCCS Humanitas Research Hospital, 20089 Milan, Italy; antonio.marzo@humanitas.it (A.M.M.); luca.dominici@humanitas.it (L.D.); ruggero.spoto@humanitas.it (R.S.); anna.bertolini@humanitas.it (A.B.); lorenzo.lofaro@humanitas.it (L.L.F.); francesco.lafauci@humanitas.it (F.L.F.); beatrice.marini@humanitas.it (B.M.); luciana.dicristina@humanitas.it (L.D.C.); marta.scorsetti@hunimed.eu (M.S.); 2Department of Radiation Oncology, Azienda Universitaria Ospedaliera Careggi, 50134 Florence, Italy; mauro.loi82@gmail.com; 3Department of Biomedical Sciences, Humanitas University, 20090 Milan, Italy

**Keywords:** lung re-irradiation, stereotactic body radiation therapy, phase I, maximum tolerated dose

## Abstract

Few data are available on the role of SBRT re-irradiation for isolated recurrences. We designed a prospective phase I study to evaluate the maximum tolerated dose (MTD) of SBRT for thoracic re-irradiation, for peripheral lung lesions. RT was delivered with a dose escalation design from 30 Gy in five fractions up to 50 Gy in five fractions. The primary end point was the definition of the maximum tolerated dose (MTD) of SBRT for thoracic re-irradiation. The dose-limiting toxicity was pneumonia ≥G3. Fifteen patients were enrolled. No cases of pneumonia ≥G3 occurred in any of our cohorts. Only one patient developed pneumonia G1 during treatment. Three patients developed acute toxicities that included dyspnea G1, cardiac failure G3, and chest wall pain. One patient developed G3 late toxicity with acute coronary syndrome. After a median follow-up of 21 months (range 3.6–29.1 months), six patients (40%) had a local relapse. Distant relapse occurred in five patients (33.3%). At the last follow-up, six patients died, all but two due to progressive disease. SBRT dose escalation for thoracic re-irradiation is an effective and well-tolerated option for patients with inoperable lung lesions after a first thoracic RT with acceptable acute and late toxicities.

## 1. Introduction

Radical dose radiotherapy (RT) is frequently used in thoracic malignancies, both for early-stage primary non-small cell lung cancer (NSCLC) and for secondary lesions from other primary tumors, particularly in the setting of oligometastatic patients. Although the development of distant metastases is the predominant pattern of failure after treatment with radical RT, isolated local recurrences are still observed and are becoming more and more frequent along with the prolongation of life expectancy in cancer patients. Salvage options for isolated post-radiation local failures are limited, with surgery or retreatment with radiotherapy as potential modalities.

While salvage surgical resection after previous thoracic irradiation has shown encouraging results, most patients experiencing post-radiation local relapse are not surgical candidates [1,2]. Therefore, re-irradiation may be the only viable salvage option for many patients. Retreatment with conventionally fractionated RT has been reported for locoregional failures, with generally poor outcomes [3,4]. Although salvage stereotactic body RT (SBRT) after initial conventionally fractionated RT (CFRT) has been described in a select small series of heterogeneous patient groups ranging from early stage to locally advanced and metastatic, few data exist to guide us on the role of re-irradiation with SBRT for isolated recurrences after initial SBRT for early-stage NSCLC [5,6,7,8]. Due to the heterogeneity and the low numerosity of these experiences, important questions regarding the safety and efficacy in this setting are still largely unanswered. Particularly, the possibility of delivering a second course of ablative dose RT (BED ≥ 100 Gy) is little explored.

Based on this background, we designed a prospective phase I study to evaluate the maximum tolerated dose (MTD) of SBRT for thoracic re-irradiation. The dose-limiting toxicity was defined as pneumonitis ≥G3.

## 2. Material and Methods

From May 2020 to October 2022, candidate patients for thoracic re-irradiation with SBRT were proposed for study enrolment. The inclusion criteria were as follows:Inoperable primary non-small cell lung cancer or other metastatic primaries with lung metastases, already treated with radical dose RT;Peripheral lesion (>2 cm from the tracheo-bronchial tree);Inoperable local recurrence (defined as a tumor recurrence overlapping the 50% isodose field) confirmed by documented radiographic findings and/or pathological biopsies within the thoracic area;Patients had previously received curative intent RT of more than 50 Gy for conventionally fractionated RT or a biologically equivalent dose of more than 75 Gy for SBRT;No active distant metastasis or controlled distant metastasis at the time of re-irradiation;Eighteen years of age or older.

Patients were treated in a supine position with a thermoplastic mask used to immobilize the thoracic region. A 4D-CT scan without a medium of contrast +/− FDG-CT/PET was performed for simulation; CT scans were acquired with a 3 mm slice thickness.

Target and critical structures were outlined for each individual patient. The gross tumor volume (GTV) was defined in any recorded respiratory phase to generate an Internal Target Volume (ITV) that accounted for patient respiratory motion. Planning target volume (PTV) was defined as the ITV plus 5 mm of margin. Critical structures were the lungs, esophagus, heart, spinal cord, large vessels, main bronchus, thoracic wall, and trachea.

The radiation treatment was delivered with three possible schedules, with a dose escalation design from 30 Gy in 5 fractions up to 50 Gy in 5 fractions, if no relevant safety alert occurred, as described in the statistical methods below. All SRT/VMAT plans were optimized by inverse planning to ensure maximal dose conformity and rapid dose falloff toward critical structures. SRT/VMAT was delivered with 6- or 10-MV photons, using modulated dynamic arcs. The dose was prescribed to the target ensuring that more than 98% of PTV will receive 95% of the prescribed dose. Patient positioning was checked with CBCT every session. Patients were evaluated on the first and the last day of therapy and more often if required.

After SRT, patients were evaluated for a follow-up within 3 months and then every 3 months for the first year, including history and physical examination, KPS, and toxicity assessment. Contrast-enhanced CT scan imaging was performed within 3 months after SRT and every 3 months thereafter. FDG-CT PET was performed, if clinically indicated, at least 6 months after radiation therapy. Hematologic and non-hematologic toxicities were graded according to Common Terminology Criteria for Adverse Events version 5.0.

Response assessment was evaluated using the RECIST criteria (version 1.1) as follows:

The trial was approved by the internal ethical committee and all patients signed a specific consent form.

The primary end point was the definition of the maximum tolerated dose (MTD) of SBRT for thoracic re-irradiation. The dose-limiting toxicity was pneumonitis ≥G3. The secondary end points were local control (LC), distant metastases-free survival (DMFS), systemic therapy-free survival (STFS) and overall survival (OS), and acute and late toxicities other than dose-limiting toxicity.

Survival times were calculated from the last day of re-SBRT until the last follow-up visit or death, whichever occurred first. Local progression was defined as a dimensional and metabolic avidity increase in the irradiated lung lesion. Distant progression was defined as the appearance of distant metastases. For STFS, events could be the initiation of a new systemic therapy, a change in ongoing therapy, an increase in the same drug dosage, or a restart of the previous systemic therapy interrupted for “drug holiday” purposes. Patients who did not start/change systemic therapy at the last follow-up were censored.

The study was designed to determine the maximum tolerated dose (MTD) that could be safely delivered in the lung re-irradiation setting. The starting dose level was 30 Gy (level 1 or L1), to be increased to 40 Gy (level 2 or L2), with a final dose of 50 Gy (level 3 or L3) in 5 fractions. The dose-limiting toxicity (DLT) was defined as grade 3 or higher pneumonitis considered to be possibly, probably, or definitely related to SBRT occurring within the first 90 days of treatment. The MTD was defined as the highest dose level in which a proportion of 1 out of 5 or fewer patients experienced DLT. Due to the late onset nature of DLT, in order to allow for continuous accrual without sacrificing patient safety or the accuracy of identifying the MTD, the Time-To-Event Bayesian Optimal Interval (TITE-BOIN) design was used [9]. Assuming a 3-level dose-escalation plan (number of patients per cohort = 5), a study sample of 15 patients was required.

Descriptive statistics were used to summarize patient-related and treatment-related variables. The Kaplan–Meier method was applied to estimate survival curves, and the log-rank test was used to compare curves of dichotomized categorical and continuous variables.

## 3. Results

Fifteen patients were ultimately enrolled. Patient characteristics are shown in Table 1.

The median patient age at the SBRT time was 72 years (range 34–86). The majority of patients (10) were retreated for a primary NSCLC. Twelve patients had previously received SBRT (48 Gy in 4 fractions in five cases, 50 to 60 Gy in 5 fractions in the remnant seven cases), two hypofractionated RTs (55 Gy in 20 fractions), and one CFRT (60 Gy in 30 fractions).

The median time interval between the first and second RT was 21.97 months (range 9.57–101.1). Histological confirmation of local relapse was obtained in three patients (20%).

Doses received by target volumes at first treatment, re-SBRT, and the sum of both courses translated in EQD2 are shown in Table 2. The median ITV volume was 12.55 cc (IQ range 18.27). The median PTV volume was 37.89 cc (IQ range 53.29). An example is shown in Figure 1.

### 3.1. Toxicity

Concerning the primary endpoint, MTD was not reached. Indeed, no cases of pneumonia ≥ G3 occurred in any of our patients’ cohorts. Only one patient developed pneumonia G1 during treatment (retreatment 50 Gy in five fractions). Three patients developed acute toxicities that included dyspnea G1 (30 Gy/5), cardiac failure G3 (40 Gy/5), and chest wall pain G1 (50 Gy/5). One patient developed G3 late toxicity with acute coronary syndrome (30 Gy/5). Neither acute nor late toxicities ≥G2 were developed by patients receiving the higher dose scheduled of 50 Gy in five fractions. No correlation between cumulative doses and toxicity was recorded, probably due to the low number of events.

Local response to re-SBRT was PR in seven patients, SD in seven patients, and PD in one patient.

### 3.2. Local Control

After a median follow-up of 21 months (range 3.6–29.1 months), six patients (40%) had a local relapse.

The median LC was 12 months (95CI 11–12). LC at 6 months, 1 year, and 2 years was 93%, 68%, and 48%, respectively (Figure 2). After univariate analysis, a dose >30 Gy was correlated with better local control (median: NR vs. 10 months, *p* = 0.045 HR 0.22 95CI 0.04–1.2) (Figure 3).

### 3.3. Survival Outcomes

Distant relapse occurred in six patients (40%). The median DMFS was 22 months (95CI 10–22). DMFS at 6 months, 1 year, and 2 years was 87%, 65%, and 49%, respectively.

Four patients started a new systemic treatment due to disease progression. The median STFS was not reached. STFS at 6 months, 1 year, and 2 years was 93%, 86%, and 63%, respectively.

At the last follow-up, six patients died, all but two due to progressive disease.

The median OS was 25 mo (95CI 22–29 mo). The OS at 6 months, 1 year, and 2 years was 100%, 92%, and 71%, respectively (Figure 4).

None of the analyzed parameters correlated with DMFS, STFS, and OS during univariate analysis.

## 4. Discussion

We report the results of a phase I trial on lung re-irradiation with SBRT dose escalation. The primary end point of the study was met since no DLT was found and a dose escalation up to 50 Gy in five fractions was safe.

To the best of our knowledge, the present study is the second prospective trial in this field and the only one specifically focused on the use of SBRT for marginal or in-field recurrences. Indeed, Sun et al. [10] enrolled 59 patients with isolated recurrent NSCLC in lung parenchyma after definitive treatment of stage I-III disease in a single-arm phase II trial. Patients were re-irradiated with 50 Gy in four fractions. However, the index lesion was distant from the prior treatment in 85% of cases, different from our study.

Indeed, we decided to include in the STRILL trial only patients in which the recurrent lung lesion was at least overlapping the 50% isodose of the previous RT. There is no uniform definition in the literature for lung re-irradiation, thus increasing the heterogeneity of available data. An Italian review [11] on this topic found that re-irradiated lesions were likely to have been fully encompassed by at least the 50% isodose of the previous radiotherapy plan and most often, wholly or partially, within the high-dose region, as in our experience. Similarly, an international Delphi consensus [12] defined lung re-irradiation as follows: “any dose of radical radiation for lung cancer, after initial radical radiotherapy to the thorax or surrounding tissues for any tumor histology, provided there is any overlap of previous dose in either the planning target volume (PTV) or the organs-at-risk (OARs)”.

Lung re-irradiation is mainly described in retrospective studies, varying from case reports to a series of hundreds of patients [13]. Selection criteria, treatment schedules, RT techniques, and data reported are highly variable; therefore, indications for clinical practice are difficult to derive. High-level evidence driving radical thoracic re-irradiation is lacking. Indeed, the Italian review cited above [11] concluded that “the frequent lack of a sufficient description of the treatment’s intent, the heterogeneity in technique and radiotherapy regimen, makes balancing risk and benefit of re-RT based on published data even more difficult”.

The main limitation for radiation oncologists is related to the risk of severe and fatal toxicities. In a recent review by Grambozov et al. [14], eleven studies were analyzed for a total number of 524 patients. Treatment-related mortality rates ranged from 2% to 14%. Acute and late toxicities higher than G3 ranged from 0 to 39% and 0 to 12%, respectively, including bleeding, radiation pneumonitis, esophagitis, and radiation-induced myelopathy. Similarly, Rulach et al. [12] reviewing the literature data reported a 5.2% to 23% risk of grade 3 pneumonitis and a grade 5 toxicity rate up to 20% depending on technique and tumor location.

SBRT in this clinical scenario is gaining appeal, due to the high precision combined with the possibility of delivering high doses per fraction; however, its feasibility and safety are not well defined yet. A paper by MD Anderson showed that patients with local recurrence after lung SBRT, re-irradiated with a second course of SBRT, obtained a significantly longer median overall survival (OS) compared to those who did not receive salvage SBRT [15]. De Ruysscher et al. [16] similarly found higher median OS rates in patients treated with SBRT compared to those irradiated with no stereotactic techniques. The same group highlighted how lethal toxicities were more common in centrally located tumors.

Wang et al. in 2023 [17] analyzed 212 patients re-irradiated with SBRT. G1-2 toxicity was reported in 164 patients [77.36%], while G3 and G4 toxicities were reported in 25 (11.79%) and 3 patients (1.42%), respectively. Three patients experienced G5 toxicity, due to bleeding and gastric perforation. Again, the authors noted a relationship between higher toxicities and the central location of the re-irradiated tumors.

In this context, our experience, thanks to its prospective design, could increase our awareness. We did not find any G3 or higher pneumonitis, and the only G3 side effects were acute coronary syndrome and cardiac failure, both in frail and comorbid patients, and for which a causal correlation with re-SBRT could be unlikely. Three patients experienced G1 toxicity, and no other side effects were recorded. The decision to exclude patients with central and ultracentral lesions from the STRILL trial obviously influenced this low toxicity rate.

The advantage of SBRT as a re-irradiation technique for lung lesions is also related to a possible higher efficacy, delivering ablative doses. Due to the heterogeneity of available data, variable outcomes can be found in the literature. In the Italian review [11], a pooled 1 yr OS rate of 60.9% and a 2 yr OS of 43.9% were reported. In the retrospective experience by Horne et al. [8], using ablative re-SBRT (median BED 106 Gy) in 72 patients with small lung recurrences, the median OS was 20.8 months. Viani et al. conducted a systematic review and meta-analysis on 20 studies and 595 patients [18], showing an estimated 2-year OS of 54%. In their experience, the interval between the first and second RT courses, tumor size, and re-irradiation dose were significantly associated with survival. Wang et al. [17] estimated 2-year OS estimates of 50%, 60%, and 70% for 41.62 Gy, 46.88 Gy, and 52.55 Gy in five fractions. In our study, outcomes in very selected patients were even better than those previously cited. Indeed, the median OS was 25 months. The OS at 6 months, 1 year, and 2 years was 100%, 92%, and 71%, respectively. Our results are in line with those reported by the other prospective experience. Indeed, Sun et al. found a 3-year OS rate of 63.5% [10].

Re-irradiation can somehow be limited in its efficacy by intrinsic disease radioresistance. Indeed, we found a median LC of 12 months with 1 and 2 years of 68% and 48%, lower than expected with primary SBRT for lung cancer or metastases. We also found a predictable correlation between re-SBRT dose and local recurrence. Indeed, patients treated with 40 or 50 Gy in five fractions had fewer local relapses than patients treated with 30 Gy in five fractions. Our results are in line with the available literature, accounting for the negative impact of patients retreated with low doses. Indeed, in a recent review, the pooled locoregional progression-free survival rate at 1 and 2 years was 65.1% and 47.2%, respectively [11].

Higher and ablative doses are probably needed to improve these results. Lee et al. [19] analyzed 20 patients retreated with SBRT, with a median dose of 54 Gy (range 48–60 Gy) in a median fraction number of 4 (range 4–6). The 1- and 2-year local control rates for the re-RT group were 73.9% and 63.3%. In the above-mentioned study by Horne et al. [8], local recurrence at 2 years was 21.6%. Lower re-irradiation BED was statistically correlated with local relapse during univariate analysis. The relevance of re-irradiation dose was also shown in the review by Grambozov et al. [14]. The median locoregional control in their analysis was 12.1 months. Again, the predicted model by Wang et al. [17] found a 2-year estimated tumor control probability of 50%, 60%, and 70% for 42.04 Gy, 47.44 Gy, and 53.32 Gy in five fractions, respectively.

We also collected, in our study, data related to subsequent therapies received by patients at relapse, with an eye on systemic treatment change or initiation. We found an STFS of 93%, 86%, and 63% at 6 months, 1 year, and 2 years. None of the published analyses on lung re-irradiation reported this outcome, to the best of our knowledge. Although STFS is commonly linked with oligometastatic disease, we feel that it can also be relevant in the re-irradiation scenario, considering that most of these patients are candidates for a new course of RT due to their intrinsic frailty and comorbidities, not allowing other local or general approaches.

We acknowledge the limitations of the present study. By including both primary and metastatic lesions, the treatment outcomes might be influenced by this histologic heterogeneity. A second source of variability is linked to the previous RT treatment since we included both patients initially treated with conventionally fractionated RT and SBRT. However, since 12 out of 15 patients were re-irradiated after a previous SBRT, we think that this limitation is partially overcome. The low number of enrolled patients, due to the study design (phase I), limits our conclusions on treatment efficacy. Even the median follow-up, needed for safety evaluation, is probably lacking the consideration of long-term outcomes and toxicity. However, the design of the study with the TITE-BOIN method allows us to partially take into account late toxicity which also uses the follow-up time to make decisions on dose escalation. We also acknowledge that patients included in the STRILL trial are a well-selected subcategory of all patients presenting with local recurrence in real clinical practice. The exclusion of central lesions surely reduced the risks of severe toxicity. At the same time, due to its prospective nature and the low-level evidence available in the literature, the STRILL trial can enlighten the safety and promising efficacy of high-dose ablative re-SBRT, at least in selected patients with small peripheral lung recurrences. This is in line with the recent expert recommendations. Rulach et al. [12] suggested the use of lung re-irradiation for “patients with a good performance status, with locally relapsed disease or second primary cancers”.

## 5. Conclusions

SBRT dose escalation for thoracic re-irradiation for peripheral lung lesions is an effective and well-tolerated option for patients with inoperable lung lesions after a first thoracic RT with acceptable acute and late toxicities. No dose-limiting toxicity was seen during the study. Anyway, a phase II trial is needed to better define the efficacy of SBRT at high doses.

## Figures and Tables

**Figure 1 diseases-12-00153-f001:**
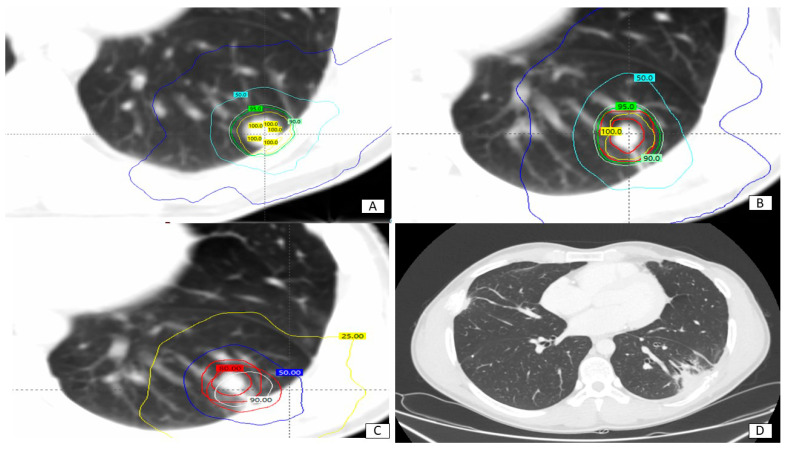
Clinical case: patient affected by lung metastases from soft tissue sarcoma, treated with SBRT (48 Gy in 4 fractions) in September 2020 (**A**), relapsing with new lower left lung nodule in May 2022(**B**), overlapping 50% isodose of previous SBRT. Retreatment in July 2022 (50 Gy in 5 fractions), with cumulative doses shown in (**C**). In (**D**), last follow-up CT scan (November 2023) shows regular appearances after SBRT and no signs of disease persistence.

**Figure 2 diseases-12-00153-f002:**
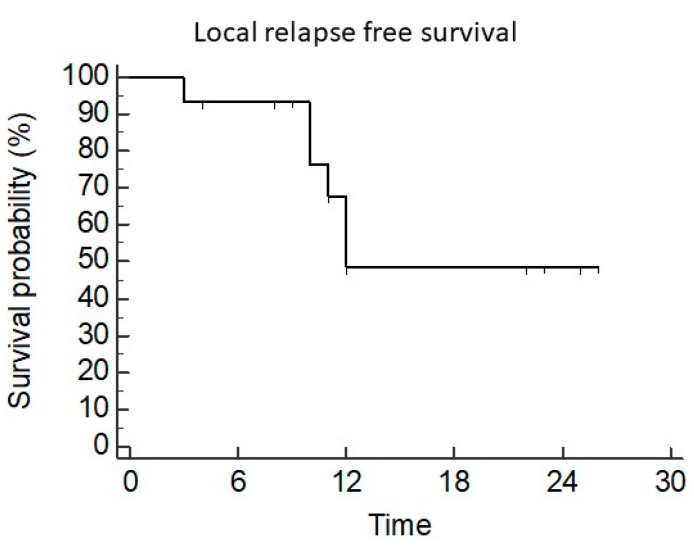
Local control.

**Figure 3 diseases-12-00153-f003:**
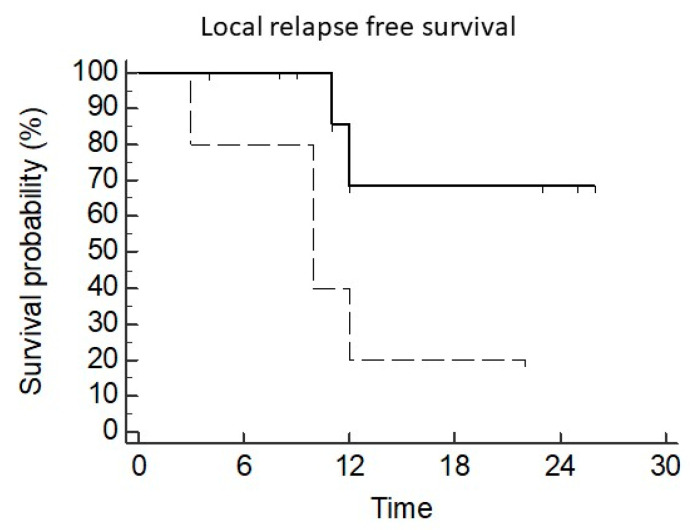
Local control according to re-SBRT dose (dashed line 30 Gy/5 fractions).

**Figure 4 diseases-12-00153-f004:**
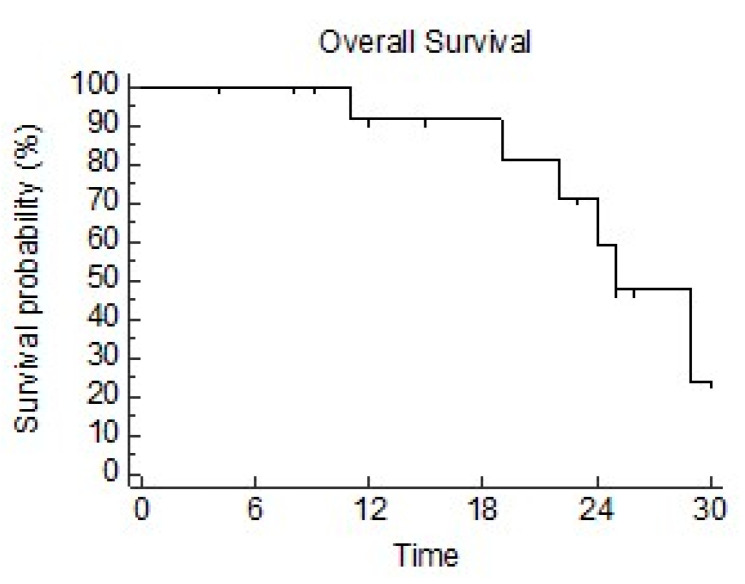
Overall survival.

**Table 1 diseases-12-00153-t001:** Patients and disease characteristics.

Features	Patients (n)	Percentage
**SEX** Male Female	11 4	73% 27%
**SMOKING HISTORY** Never Former Current	2 9 4	13% 60% 27%
**COMORBIDITIES** 0 1 >1	3 11 1	20% 73% 7%
**PULMONARY COMORBIDITIES** No BPCO	13 2	87% 13%
**CARDIAC COMORBIDITIES** No Hypertension Atrial fibrillation Heart failure Acute coronary Syndrome	5 8 3 1 1	33% 53% 20% 7% 7%
**PRIMARY CANCER SITE** Lung Larynx Rectum Sarcoma Unknown	10 2 1 1 1	66% 13% 7% 7% 7%
**PRIMARY CANCER HISTOLOGY** Adenocarcinoma Squamocellular carcinoma Synovial sarcoma Epithelial NOS	8 5 1 1	53% 33% 7% 7%
**PREVIOUS CANCER TREATMENTS** Surgery Systemic Therapy	6 8	40% 53%
**FIRST RT ON TARGET LESION TYPE** SBRT Hypofractionated RT Conventional RT	12 2 1	80% 13% 7%
**OVERLAP** In site 50%	6 9	40% 60%
**PS AT SBRT** 0 1	9 6	60% 40%
**SBRT TOTAL DOSE IN 5 FRACTIONS** 30 Gy 40 Gy 50 Gy	5 5 5	33% 33% 33%

**Table 2 diseases-12-00153-t002:** Doses received by target volumes at first treatment, re-SBRT, and the sum of both courses translated in EQD2, α/β 3 for OARs, α/β 10 for tumor.

Volume	First RT Course Dose (Gy) Median; Range; IQ Range	Re-SBRT Dose (Gy) Median; Range; IQ Range	Sum in EQD2 (Gy) Median; Range; IQ Range
PTV Dmax	54.6; 51.12–68.92; 6.855	42.5; 31.6–53.17; 18.21	148.7; 104.3–199.7; 47.75
PTV Dmean	40.2; 18.28–54.18; 16.345	40; 29.86–50.45; 20	120; 75.4–171.30; 34.9
Bronchus Dmax	2.73; 0.13–57.71; 31.385	8.05; 0.13–20.11; 12.965	9.2; 0.2–101.2; 76.6
Esophagus Dmean	8.16; 5.14–57.29; 13.46	7.6; 5.6–12.02; 3.49	18.2; 11–71.7; 33.6
Spinal cord Dmax	8.62; 0.25–29.14; 13.215	5.59; 0.32–14.05; 4.16	13; 3.1–66.4; 14.85
Large vessels Dmax	9.09; 0.4–58.26; 32.885	4.33; 0.16–18.23; 8.43	10.1; 0.2–107.1; 58.1
Lung Dmean	2.91; 1.11–12.50; 4.355	2.24; 0.92–5.21; 1.155	3.9; 1.3–10.5; 5.7
Heart Dmax	7.74; 0.2–58.1; 21.93	7.46; 0.17–47.15; 14.62	15.99; 0.2–151.2; 69.94

## Data Availability

Research data are stored in an institutional repository and will be shared upon request to the corresponding author.

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
