# Peer review of "STRILL: Phase I Trial Evaluating Stereotactic Body Radiotherapy (SBRT) Dose Escalation for Re-Irradiation of Inoperable Peripheral Lung Lesions"

_diseases, 2024, doi:10.3390/diseases12070153_

Round 1

Reviewer 1 Report

Comments and Suggestions for Authors

Overall, the manuscript was prepared relatively well and contains information that could interest many readers. There are few comments for improvement.

Major

It seems critical to explicitly indicate in title, abstract, and conclusion that centrally located legions were excluded in this trial. 

Minor

Line 171: "was RP in" => "was PR in"?

Figure 2 and 3: "Local Relapse" => "Local Relapse Free"? 

Line 255: "OS rate was 60.9%, 2-yr OS was 43.9% were" => "OS rate of 60.9%, 2-yr OS of 43.9% were"?

Author Response

Comment 1: It seems critical to explicitly indicate in title, abstract, and conclusion that centrally located legions were excluded in this trial. 

Response 1: done in title, abstract and conclusion

Comment 2: Line 171: "was RP in" => "was PR in"?

Response 2: yes, correct in the revised manuscript

Comment 3: Figure 2 and 3: "Local Relapse" => "Local Relapse Free"? 

Response 3: changed in revised manuscript

Comment 4: Line 255: "OS rate was 60.9%, 2-yr OS was 43.9% were" => "OS rate of 60.9%, 2-yr OS of 43.9% were"?

Response 4: changed in revised manuscript

Reviewer 2 Report

Comments and Suggestions for Authors

The paper titled by “STRILL: A Phase I Trial Evaluating STereotactic body Radiotherapy (SBRT) Dose Escalation for reIrradiation of Inoperable Lung Lesionsby Franceschini et al. is a phase I dose escalation study, showing that SBRT with ablative purpose can be safely delivered for recurrent lung lesions after previous radical radiotherapy. The paper is well written with an extensive and highly detailed background. It flows logically.  

Some comments: 

  • -Table 1, please add percentage 

  • -Please, add the previous cancer treatments that the patients received 

  • -Please, add up to date literature 

Author Response

Comment 1: Table 1, please add percentage 

Response 1: done

Comment 2: add the previous cancer treatments that the patients received 

Response 2: added in table 1

Comment 3: Please, add up to date literature 

Response 3: frankly, we think that the literature is update and logical with the discussion we made. Since this is note a review of the literature, we acknowledge that literature will be incomplete. If you have specific suggestions, please tell us, we will be glad to add them in the discussion if deemed of added value to what already reported

Reviewer 3 Report

Comments and Suggestions for Authors

Paper deal about re-irradiation of recurrent lung cancer not suitable for surgical resection. Despite the sample size is not so large the study is prospective and correct from methods point of view. Results are clearly reported and support very well the conclusions of the paper. The topic is not widely investigated in the literature morover is actual and interesting for the general audience. I have not substantial critics to be advanced on this manuscript and I suggest the paper should be considered for the publication in the present form

Comments on the Quality of English Language

English language is understandable and readable requiring only minor adjustments.

Author Response

Comment 1: Paper deal about re-irradiation of recurrent lung cancer not suitable for surgical resection. Despite the sample size is not so large the study is prospective and correct from methods point of view. Results are clearly reported and support very well the conclusions of the paper. The topic is not widely investigated in the literature morover is actual and interesting for the general audience. I have not substantial critics to be advanced on this manuscript and I suggest the paper should be considered for the publication in the present form

Response : thanks!